# The Effect of Smoking on the Number and Type of Rectal Aberrant Crypt Foci (ACF)—First Identifiable Precursors of Colorectal Cancer (CRC)

**DOI:** 10.3390/jcm10010055

**Published:** 2020-12-26

**Authors:** Marek Kowalczyk, Łukasz Klepacki, Ewa Zieliński, Waldemar Kurpiewski, Krzysztof Zinkiewicz, Łukasz Dyśko, Wiesław Pesta

**Affiliations:** 1Department of Psychology and Sociology of Health and Public Health, University of Warmia and Mazury, 10-082 Olsztyn, Poland; forel@neostrada.pl; 2Clinic of Oncological and General Surgery, University Clinical Hospital in Olsztyn, 10-082 Olsztyn, Poland; lklep31@gmail.com (Ł.K.); wkurpiewski@interia.pl (W.K.); lukaszdysko@gmail.com (Ł.D.); wieslaw.pesta@vp.pl (W.P.); 3Department of Anatomy, University of Warmia and Mazury in Olsztyn, 10-082 Olsztyn, Poland; 4Department of Emergency Medicine and Disaster Collegium Medicum in Bydgoszcz, Nicolaus Copernicus University, 87-100 Toruń, Poland; 52nd Department of General, Gastroenterologic and Gastrointestinal Oncologic Surgery, Medical University of Lublin, University Hospital No.1, 20-059 Lublin, Poland; kzinek@yahoo.com

**Keywords:** rectal aberrant crypt foci (ACF), colorectal cancer (CRC), smoking

## Abstract

Background: The problem of smoking and its influence on the occurrence of precursors and advanced colorectal cancer is often discussed in the medical literature. Tobacco smoke can provide a non-nutritional source of polycyclic hydrocarbons and other substances which, when combined with an incorrect diet, may play a role in promoting carcinogenesis at the level of the genetic control mechanism. The aim of our study was to assess the effect of smoking on the frequency and type of aberrant crypt foci (ACF) in the rectum and polyps in the large intestine in people who smoke more than 20 cigarettes a day for more than 20 years. Methods and Findings: A colonoscopy combined with rectal mucosa staining with 0.25% methylene blue was performed in 131 patients. Each of the study participants gave informed consent to participate in the study. Three bioptates were collected from the foci defined macroscopically as ACF; in cases where there were fewer foci, the number of collected foci was respectively lower. On the colonoscopy day, patients completed the questionnaire regarding epidemiological data used for analysis of factors affecting the occurrence of ACF in the study group. The number of ACF in the colon was divided into three groups: −ACF < 5, 5 < ACF < 10, ACF > 10. In the statistical analysis, numerical data were presented and real numbers, range of arrhythmic means, mean standard deviation, and results of probability distribution. The Student’s test, U test, and chi2 were applied in order to determine the significance of differences of means and frequency of events in both groups. The level of significance was set at α = 0.05. Statistica 7.1 and Excel 2010 were used. Most smokers are in the age groups between 51–70 years. In the youngest (31–40 years), single ACF appear first (ACF <5) ACF in the number of 5–10 appear a little later (around 50 years of age) and dynamically increase, reaching a maximum at the age of 60–65.ACF in the number >10 appear at the latest age (55 years old) and their number gradually increases with age (linear growth). The probability of occurrence of ACF in all groups is greater in smokers, and the difference for the ACF group 5-10 and ACF >10 is statistically significant with a significance level of *p* < 0.05. Apart from ACF normal, all types of ACF are more likely in this group than in non-smokers and these differences are statistically significant with *p* < 0.05. Conclusions: Smoking has a significant impact on the number and type of rectal ACF. Smokers have a greater number of ACFs in the rectum than non-smokers, and the most common type is hyperplastic ACF. Smokers are more likely to develop polyps in all sections of the colon compared to non-smokers.

## 1. Introduction

Numerous publications deal with the problem of stimulants, including smoking, and their impact on the occurrence of benign and malignant neoplastic lesions in the large intestine [1].

The process of developing colorectal cancer is preceded by indirect changes, only a small part of which undergoes malignant transformation. The first identifiable change in CRC carcinogenesis is aberrant crypt foci (ACF), which is identified by colonoscopy using indigo carmine or methylene blue [2].

Since it is recognized that ACF may a precursor of adenomas, other types of polyps and CRC can still be affected by toxins released during smoking. The relative risk of toxins related changes depends on the number of cigarettes smoked per day and duration of smoking. The ingredients of tobacco smoke in combination with an incorrect diet increase the risk of CRC formation [3].

According Samowitz et al. [4], tobacco increases the risk of colon cancer by abnormal methylation of CpG islands and by the impact of the instability of the genome.

According to Anderson et al. [5], smokers have a higher ACF count in the rectum and sigmoid colon than non-smokers. The right-sided location of CRC is more common, which is consistent with the assumption that the right-sided location of colorectal cancer is more often associated with microsatellite instability (MSI) disorders, which we find more often in smokers than non-smokers. Researchers suggest that smoking plays a particularly important role in the MSI-dependent carcinogenesis pathway, where the initial stage is hyperplastic ACF and the intermediate form is serrated adenoma. This is the second of the hypothetical pathways of carcinogenesis in CRC proposed by Jass et al. It is recognized that 15–25% of CRC may be formed in this way, especially located in the right half of the colon [6,7].

Aim: The aim of our work (study) was to assess the impact of smoking on the frequency and type of ACF in the rectum and colon polyps in people who smoke more than 20 cigarettes a day for more than 20 years.

## 2. Methods

The project was approved by the Bioethics Committee at the Faculty of Medical Sciences of the University of Warmia and Mazury in Olsztyn—Resolution No. 11/2010. A colonoscopy combined with rectal mucosa staining with 0.25% methylene blue was performed in 131 patients. Each of the study participants gave informed consent to participate in the study. The patients were divided into 2 groups- smokers and non-smokers. Smokers—a person who regularly smokes 20 cigarettes a day for 20 years (20 pack/years). Non-smokers-person who has never smoked cigarettes. Three samples were collected from the foci defined macroscopically as ACF; in cases where there were fewer foci, the number of collected foci was respectively lower. On the colonoscopy day, patients completed the questionnaire regarding epidemiological data used for analysis of factors affecting the occurrence of ACF in the study group. The number of ACF in the colon was divided into three groups: – ACF < 5, 5 < ACF < 10, ACF > 10. Colonoscope CF-Q-165 L, “biopsy forceps” FB-240 U Olympus and catheter type spray Olympus company were used in the examination. All 131 subjects underwent a full colonoscopy. After routine colonoscopy, the rectal mucosa was stained with 0.25% solution of methylene blue from the serratus line to the medial Houston’s valve. ACF were assessed using the endoscopic criteria established by Roncucci [8]. In the statistical analysis, numerical data were presented and real numbers, range of arrhythmic means, mean standard deviation and results of probability distribution. The Student’s test, U test, and chi2 were applied in order to determine the significance of differences of means and frequency of events in both groups.

The level of significance was set at α = 0.05. Statistica 7.1 and Excel 2010 were used.

## 3. Results

The study group consisted of 73 women and 58 men. ACF in the rectum was found in 119 people out of all 131. The mean age in the study group was 67 years for women and 52 years for men. The age distribution in the groups was found “normal”, which enabled the evaluation of ACF incidence and characteristics according to age. The incidence of ACF in the study population correlates with the incidence of CRC in the entire population.

Numeric data are presented in Table 1:

In the youngest group (31–40 years), single ACFs appear first (ACF < 5). From the age of 40, a gradual increase in this numerical range of ACF has been observed, which has been following a downward trend after around 60 years of age.

ACF in the number of 5–10 appear a little later (around 50 years of age) and dynamically increase, reaching a maximum at the age of 60–65, after which their growth is proportional.

ACF in the number >10 appear at the latest age (55 years old) and their number gradually increases with age (linear growth), reaching a maximum quantity in patients approximately 75 years old.

Most smokers are in the age groups between 51–70 years.

The probability of occurrence of ACF in all groups is greater in smokers, and the difference for the ACF group 5–10 and ACF >10 is statistically significant with a significance level of *p* < 0.05.

Probability distributions are similar in both groups.

The distribution of ACF types in non-smokers is exponential.

The maximum relative frequency (rf) for this group = 0.62 is for ACF normal, which occur with greater probability compared to other types of ACF (also in aggregate) and the difference of these probabilities is statistically significant with *p* < 0.01.

The distribution of ACF types in smokers is similar to normal.

Apart from ACF normal, all types of ACF are more likely in this group than in non-smokers and these differences are statistically significant with *p* < 0.05.

In this group, ACF hyperplastic is most likely to occur with a maximum of rf = 0.42.

Smokers are more likely to have polyps in all sections of the colon than smokers.

The dynamics of polyp growth in smokers and non-smokers in the right half of the colon (segment of colon—1, 2/3) is similar and is exponential with a high indicator of determination.

Smokers: Y = 0.054e ^0,4056x^ R^2^ = 0.9369

Non-smokers: Y = 0.0087e ^0,8524x^ R^2^ = 0.9885

Distally from the splenic flexion of the colon, the nature of the increase in the number of polyps changes to logarithmic (segment of colon—2/3, 4).

For smokers y = 0.2024lnx-0.0316 and a high determination index R^2^ = 0.9988

For non-smokers y = 0.2006lnx-0.0978 and high determination rate R^2^ = 0.9971

The growth decreases with a continuous increase in the number of polyps.

The relative probability of developing a type of polyp in both groups is similar and the differences are not statistically significant. In both groups, the hyperplastic polyp is most likely to occur.

In the group of people with ACF < 5 for smokers there are mainly small and medium adenomas (<20 mm), and for non-smokers only small (<10 mm).

In the group of people with ACF 5–10 for smokers there are all sizes of adenomas and for non-smokers only small and medium-sized (<20 mm).

In the group of people with ACF >10 medium and large adenomas are more likely for smokers than for non-smokers, and this difference is statistically significant with a significance level of *p* < 0.05.

## 4. Discussion

In the study group, smokers for more than 20 years had a greater number of rectal ACFs than non-smokers regardless of the number of ACFs found. This difference is statistically significant especially in the group of people with ACF 5–10 and ACF >10. If we follow the dynamics of ACF growth in the rectum in different age groups, we can see that in the youngest individuals the first appear single ACF (ACF <5). From the age of 38, a gradual increase in this numerical range of ACF is observed, and from around 60 years of age it will follow a downward trend. ACF >10 appear the latest (age 55) and their number increases gradually with age (linear increase). They reach their quantitative maximum at the age of 77 (Figure 1). The largest differences in the number of ACF in smokers vs non-smokers can be seen in the 41–50 and 51–60 age groups. In the smokers group the highest probability of ACF number 5–10 is in the age group 50–65 years, while for non-smokers in the age group 60–75 years. The probability distributions of the amount of ACF in both groups (smokers/no-smokers) are similar, which may suggest that the predisposition factors for the occurrence of ACF in both groups are also similar (Figure 2). In the smokers group there is an additional, significant mutagenic factor (tobacco smoke components) that can have a large impact on the occurrence of more ACF in the smokers group. In the study of Moxon et al. [9] researchers report the largest difference in the number of ACF for smokers in the 39–60 age group and for non-smokers in the 60–70 age group. In addition, researchers have shown that the incidence of ACF in non-smokers was strongly age dependent.

A similar relationships can be seen in the study group, especially in the 41–70 age range (Figure 3).

If we notice a global tendency to lower the minimum age to 15–19 years at which people start smoking, we can expect the occurrence of ACF, polyps and CRC in younger people.

This is a big epidemiological problem, because as we know the carcinogenesis process in CRC takes a few to several years. People who start smoking at such a young age (15–19) are at risk of symptomatic CRC younger than sporadic CRC in non-smokers. According to the report covering the period 1980–2012 published in JAMA, we can see that the number of cigarettes smoked worldwide increased by 26% in the same period under assessment. Only in four countries has it been possible to reduce smoking by more than 50%, both among men and women [10].

In the article published in 2010, Anderson et al. (5) report a greater number of ACFs found in the rectum and distal sigmoid colon ACF (>15) in smokers (20 pack years). Moreover, the authors report that people who have stopped smoking have a higher risk of ACF than non-smokers. Although in this group ACF were determined up to 20 cm from the toothed line in the rectum (larger area studied than in our study), these data confirm our results. Other researchers also report that smokers have a greater number of ACFs in the rectum than non-smokers [9,11,12].

However, there are studies that do not confirm the presence of a greater number of ACF in smokers [13]. There was also no effect of smoking period on rectal ACF in this study.

Prolonged expression to the components of tobacco smoke can leave permanent changes in the genetic material of intestinal epithelial cells. The mechanism of defense against the effects of a chemically aggressive environment depends on the rapid renewal of colon epithelial cells. As a result, new cells constantly replacing the “old-used” liquidated in the process of apoptosis, and this is reduced especially in smokers [14]. In the event of loss of control over cell renewal, mutated cell clones may form which, under favorable conditions, may give rise to CRC. This hypothesis is consistent with the data provided by D. Limsui et al. [14]. In their study, they showed that epigenetic modification may be functionally involved in colorectal carcinogenesis associated with cigarette smoking, which is particularly evident in the case of CRC: MSI-H, CIMP + and BRAF +. Cigarette smoke contains many components, at least several dozen of which are carcinogens and can significantly affect DNA methylation disorders, direct damage to the DNA helix, transcription process (increase in intracellular calcium level), and the regulation of the expression of methionine adonosyltansferase 2A, and by this enzyme the disorder of DNA methylation alone (as a result of cellular hypoxia caused by carbon monoxide contained in cigarette smoke) [15]. Nicotine changes the perception of taste in smokers. This can change the nutritional pattern towards saturated fat and salt. In addition, the diet of smokers has an insufficient share of fresh fruit and vegetables, vegetable fats, milk and other dairy products. Smokers in excess reach for food products such as red meat, fried and grilled dishes, saturated animal fats, coffee, and alcoholic beverages, which in combination with tobacco smoke nitrosamines gives a greater risk of damage to the genetic material of intestinal epithelial cells [15,16].

It should be remembered that the components of tobacco smoke enter the intestine not only through the bloodstream absorbed in the lungs, but also through the skin or directly as a result of swallowing. It is multi-site and long-lasting effect. Smokers have also been found to reduce selenium levels, which plays a role in cancer prevention (including CRC) by activating glutathione peroxidase, the responsible enzyme among others for protecting cellular DNA against oxidative damage [17].

The generation of free radicals in the intestinal epithelium under the influence of tobacco smoke can also be enhanced by the hydroxyl radical formed in the large intestine by the Fenton reaction, which additionally creates the possibility of DNA damage [18].

Perhaps the gradual thickening of the stool from the cecum to the rectum causes thickening of compounds that damage enterocytes. On the contrary, it can be caused by a diet with a high content of plant fibers, their presence in the stool shortens the time of intestinal transit, which in turn shortens the contact time of the nutrient content with the resorbing surface of the intestine. In this way, contact between carcinogens and intestinal mucosa cells is limited [3,19].

The second beneficial mechanism of action of dietary fiber on the large intestine is to increase the mass of feces and to bind cholesterol and bile acids, thus preventing their transformation into other compounds, also of a carcinogenic nature and by increasing the amount of butyric acid derivatives which protect colonocytes [20,21,22].

The abnormal gut flora may increase the amount of secondary bile acids which are pro-carcinogenic [23].

Among the intestinal bacteria, other bacteria and yeasts that have beneficial effects in the digestive process as well as in maintaining the correct balance among the bacterial flora of the large intestine deserve attention.

In the research of Hong B et al. [24] the previously postulated effect of some bacteria (Fusobacterium nucleatum) on stimulating the growth of early CRC or on reducing local inflammation and protective effect on the epithelium of the large intestine (Faecalibacterium prausnitzii) was confirmed. Hong et al. showed that normal mucosa is more homogeneous than ACF mucosa in terms of bacterial diversity. The results of this study suggest that colon bacterial culture colonies in people with ACF are significantly different compared to people without ACF (with normal mucosa).

Dysbiosis of the microbiome of the colon may partially contribute to the development of more advanced forms of colorectal cancer.

Assessing the effect of tobacco in the presence of ACF and polyps in the large intestine should be carried out in conjunction with determining the type of lifestyle and diet, as well as factors that have a significant effect on the number of ACF and colonic polyps [3,12,25].

Do the components of tobacco smoke only affect the number of ACFs, and maybe also their type?

In the research of Anderson et al. [5] smokers were more likely to have a higher serrated ACF than non-smokers, which was in line with previous studies that suggested that smokers have a higher risk of serrated polyps [26], which may be hyperplastic ACF precursors.

In our study, most smokers were found to have hyperplastic ACF in the rectum, while patients not smokers most often stated ACF normal (Figure 4). Moreover, in smokers all kinds of ACF appeared in greater numbers in addition to normal ACF, and these differences were statistically significant (Figure 5). In the study of Muth et al. [11], the correlation between the occurrence of a higher number of ACF in smokers was confirmed, and in this study group it was found that 98% of ACF is hyperplastic and only 2% dysplastic.

One study found that smokers with a shorter smoking period have a positive correlation with small adenomas, which may suggest that the mutagenic effect of tobacco smoke components may be of particular relevance in the early periods of carcinogenesis. [19,26] Our study shows that adenomas >10 mm were more common in smokers, while non-smokers were more often found to have adenomas <20 mm (Figure 6). Similar data is provided by Moxon et al. [9].

Their study showed that smokers are more likely to have adenomas than non-smokers and have more ACF in people with advanced cancer (Carcinoma in situ and CRC), as confirmed by previous Japanese studies Takyama et al. [27].

In the study of Figueiredo et al. [28], a significant relationship was found between smoking and the occurrence of serrated polyps, which was not confirmed for traditional adenomas. In addition, serrated polyps were more common in the left half of the colon than in the right, as confirmed by previous studies [29].

In the research of Paskett et al. [30], a higher number of hyperplastic polyps was found in the distal rather than the proximal segment in smokers. In addition, researchers have found that smoking does not significantly affect the occurrence of more adenomas in smokers. Similar data are presented by Morimoto et al. [31].

In the study group smokers are more likely to have polyps in all sections of the large intestine compared to non-smokers. The growth dynamics of polyps in the right colon was greater than in the left half, but it was in the left half of the colon that more polyps were found. In both groups, hyperplastic polyps were most common, but more often in smokers with a statistically significant difference (*p* < 0.05) (Figure 7).

Distally from the splenic flexion of the colon, the nature of the increase in the number of polyps, changes from exponential (right half) to logarithmic. Growth decreases, but the number of polyps increases. Why do smokers distally from the spleen flexion have more polyps? Probably other factors besides tobacco smoke also have an impact on this. According to many authors, tumors with proximal localization in the large intestine are more often associated with microsatellite instability (MSI) disorders, and smoking plays a particularly important role in the pathway of MSI-dependent carcinogenesis, where the hyperplastic ACF is the initial stage and the serrated adenoma is the intermediate form. The components of tobacco smoke also increase the risk of developing colorectal cancer through disorders of CpG island methylation and by affecting genome instability. In colon cancers in smokers of over 20 cigarettes a day, genetic disorders such as microsatellite instability, k-ras and BRAF mutations are more common. [4,14,32].

Roncucci et al. [33] suggest that up to 50 years of age, the primary growth process for ACF is an increase in cellular proliferation, and above 50 years of age exposure to carcinogens and reduction of apoptosis may be of major importance. In our study, this can be seen significantly in Figure 7, where the number of polyps particularly dynamic increases in the right half of the colon, and much less dynamically than the splenic flexion distally. Probably in the right half of the colon, the dynamics of tumor cell growth in polyps is dependent on DNA metalization and microsatellite instability (MSI), which is particularly affected by tobacco smoke components, and distally from the splenic flexion probably the effects of other carcinogens accumulated in the distal colon are more likely. Both factors (increasing cell proliferation and reducing apoptosis), apart from age, play a key role in the development of ACF and colorectal cancer.

Studies are available that indicate that smokers have a higher risk of developing colorectal cancer in the sigmoid colon and rectum [14,34].

What are the genetic disorders in left-sided tumors? In the description of various researchers, they are similar to the right-sided, however, with a lower frequency of disorders in the MMR and MSI and BRAF genes.

Some authors suggest that in addition to the effects of tobacco smoke, other nutrients, the composition of bacterial flora and immune response cells may affect this [2,32,35,36].

We find works in which the authors also suggest that embryological development and related vasculature and innervation may play a role in carcinogenesis [37,38].

As we know, from another section of “intestine in embryonic development“, the right (midgut) and left (hindgut) half of the colon develops. The proximal segment of the colon maintains a higher concentration of short chain fatty acids and alcohols and the products of protein metabolism predominate in the distal colon. In the right half of the colon, neutral mucopolysaccharides in the mucus predominate, while in the rectum, acid mucins predominate. The two sections of the colon also differ in the number and quality of bacterial cultures, which through fermentation products can detach a significant effect on enterocyte metabolism. Depending on the type of fermented substrate, intestinal bacterial cultures and the type of mucin, the pH inside the large intestine may change locally, which may promote (alkaline pH) or limit (acidic pH) toxic and carcinogenic effects of various chemical compounds present in feces [3,37,38].

An interesting suggestion was developed by Luchtenborg et al. [39], who found differences in the occurrence of CRC in the colon and rectum depending on the type of cigarettes smoked. Smokers of non-filter cigarettes were found to have a higher incidence of colon cancer, while rectal cancer rate was higher in smokers of cigarettes with a filter.

Much work has been devoted to COX-2 induction by tobacco smoke components. Activated COX-2 generates excess TXA2 and PGE2. TXA2 can generate mitogenic factors, promote angiogenesis, activate platelets and inhibit the process of apoptosis. In addition, excessive release of PGE2 from fibroblasts, monocytes, neutrophils or dendritic cells has an immunosuppressive effect, increases the angiogenesis associated with the presence of a tumor and inhibits the process of apoptosis [40,41].

In the study of Figueiredo et al. [42] greater COX-2 overexpression in smokers compared to non-smokers with a difference of *p* = 0.02, and overexpression of COX-2 in ACF foci compared to normal mucosa in the ACF environment in smokers (*p* = 0.03).

Summing up, we see how multi-directional and multi-site the action of tobacco smoke components is, which can lead to irreversible damage to the genetic material of colonocytes and, as a result, to the formation of ACF, adenomas (serrated lesions), and CRC. Many studies show that smoking time and the number of cigarettes smoked are known as pack-years [5,9,14,19,43,44].

It should also be remembered that it is possible to transform ACF in different sections of the colon into adenomas, hyperlpastic polyps or “serrated polyps” [6,45,46,47,48].

If we recognize that smoking increases the amount of rectal ACF, especially hyperplastic and dysplastic, and that it correlates with the presence of “serrated” lesions, adenomas and CRC in the right-sided location, these people can be included in the high-risk group and subject to special endoscopic supervision.

## 5. Conclusions

Smoking has a significant impact on the number and type of rectal ACFs.

Smokers have a greater number of ACFs in the rectum than non-smokers, and the most common type is hyperplastic ACF. Smokers are more likely to develop polyps in all sections of the colon compared to nonsmokers.

## Figures and Tables

**Figure 1 jcm-10-00055-f001:**
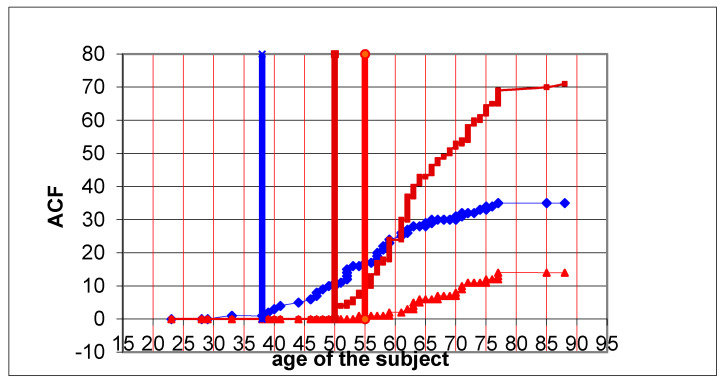
Prevalence of aberrant crypt foci (ACF) in age groups. Blue line: ACF < 5; brown line: ACF 5–10; red line: ACF > 10 0.

**Figure 2 jcm-10-00055-f002:**
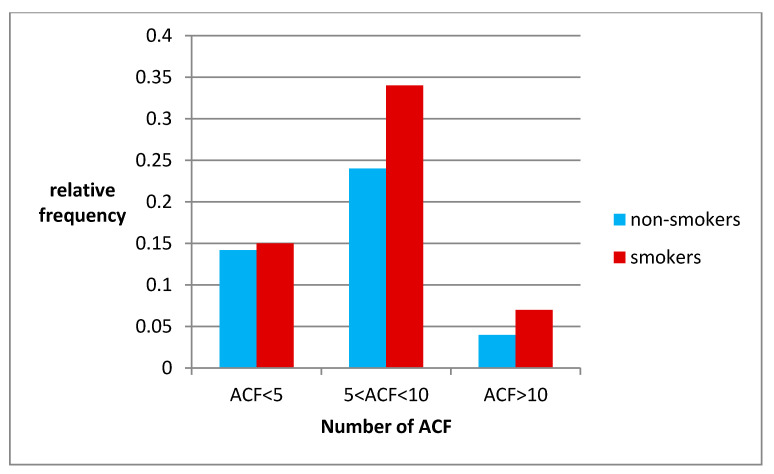
Relative probability of ACF number in the study group.

**Figure 3 jcm-10-00055-f003:**
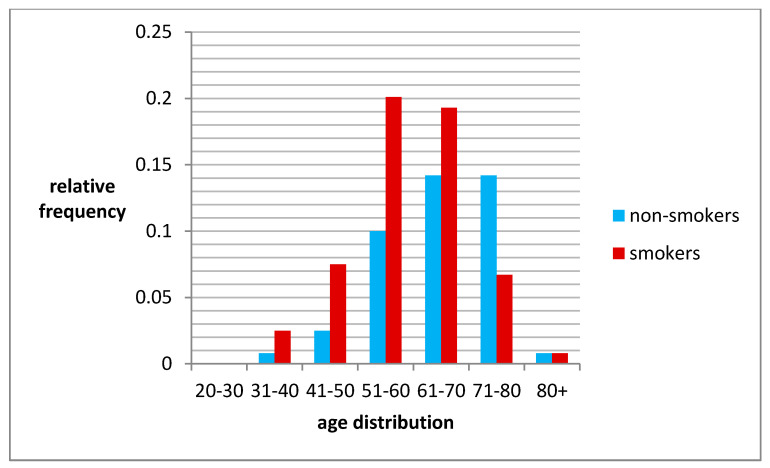
Occurrence of smokers/non-smokers in age groups.

**Figure 4 jcm-10-00055-f004:**
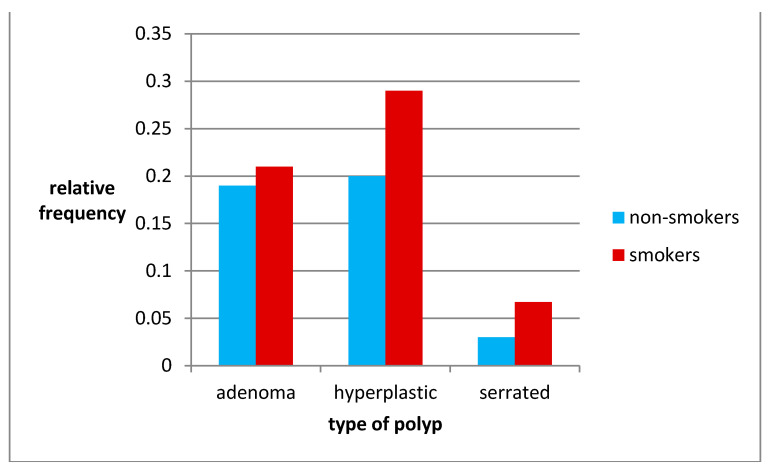
Occurrence of polyp types in smokers and non-smokers.

**Figure 5 jcm-10-00055-f005:**
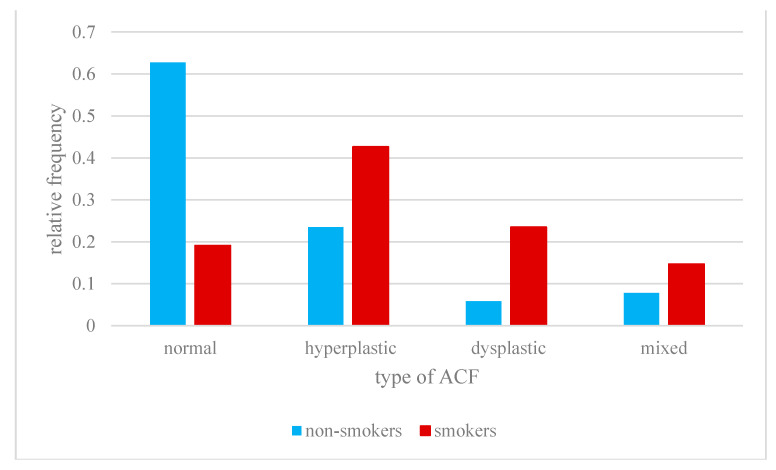
Occurrence of ACF types in the study group.

**Figure 6 jcm-10-00055-f006:**
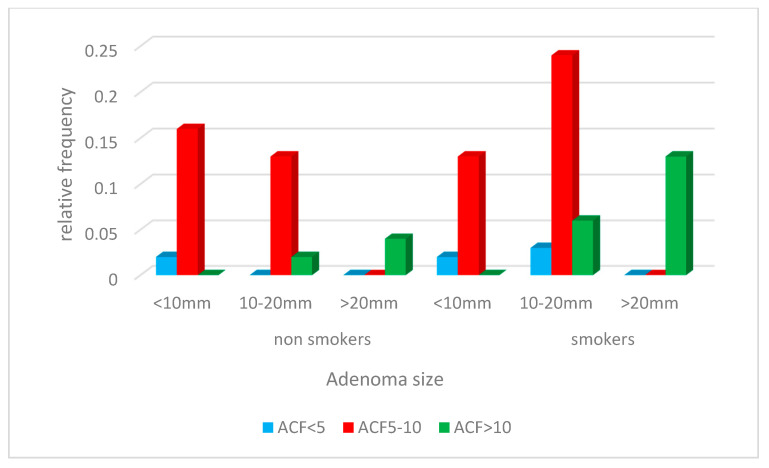
Adenoma size depending on the rectal ACF number in the smokers group.

**Figure 7 jcm-10-00055-f007:**
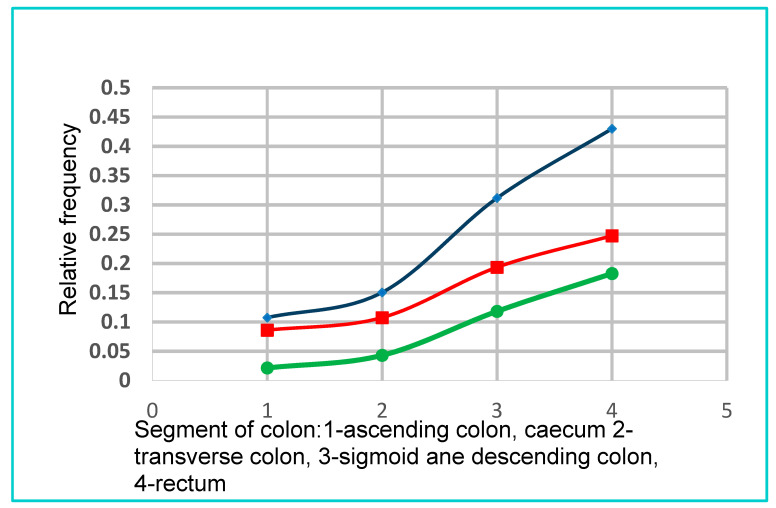
The presence of polyps in the study group in different sections of the colon. Green line: non-smokers; red line: smokers; blue line: cumulative curve.

**Table 1 jcm-10-00055-t001:** Numeric data.

		Non-Smokers	Smokers
Number of ACF	<5	17	18
	5–10	29	41
	>10	5	9
Type of ACF	Normal	32	13
	Hyperplastic	12	29
	Dysplastic	3	16
	Mixed	4	10
Type of polyp	Adenoma	14	25
	Hyperplastic	19	30
	Serrated	4	8
Adenoma (size)	<10 mm	1	2
	10–20 mm	14	19
	>20 mm	3	9
Age groups of subjects (years old)	<30	0	0
	31–40	3	1
	41–50	9	3
	51–60	24	12
	61–70	23	17
	71–80	8	17
	>80	1	1
location of polyps in different sections of the large intestine	rectum	14	16
	sigmoid and descending colon	8	123
	transverse colon	6	12
	ascending colon, caecum	3	9

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
