# Peer review of "The Effect of Smoking on the Number and Type of Rectal Aberrant Crypt Foci (ACF)—First Identifiable Precursors of Colorectal Cancer (CRC)"

_jcm, 2020, doi:10.3390/jcm10010055_

Round 1

Reviewer 1 Report

Very interesting manucript

Very interesting manuscript describing the methods used in animal models applied on humans.

The manuscript however lack a fully description of the patient group used in this trial, eg. Why have the  patients been invited for a colonoscopy? The range of age seems to be a problem in the study. In addition, the definition of smokers and non-smokers have to be described in the M&M section.

Furthermore, a clearly description of which part of the intestine has been investigated for polyps compared to the findings of ACFs in the rectal part. This should also be outlined in table 1 and maybe values of variations should be added.

The numbers of small polyps < 10 mm are very low – is this number correct? Any cancers?

The discussion is far too long and have to been compressed and shortened. There are too many sections in the discussion. The timeline in development of small ACF to large SCF to small adenomas to large adenomas should be discussed further.

Author Response

Our responses are presented in the file in the attachement

Reviewer 2 Report

The English needs to be greatly improved and edited. Far too many typos and grammatical errors. (e.g., bacground, bioptates)

What does this phrase in the abstract actually mean - "may play a role in promoting carcinogenesis at the level of the genetic control mechanism" -  the level of genetic control mechanism? In fact, this sentence in the intro does not make sense: Tobacco smoke can be a beyond nutritional source of polycyclic aromatic hydrocarbons and other substances that, in combination with an incorrect diet, can play a role in carcinogenesis at the level of the genetic control mechanism. (3)

Please provide the current PubMed citation for this paper: Anderson JC, Pleau DC, Rajan TV Increased frequency of serrated aberrant crypt foci among smokers 390 The American Journal of Gastroenterology advance online publication, 16 march 2010

Reference 18 is not appropriate for the discussion of damage to colonocyte organelle DNA and proteins. It is from 1989 and is not relevant. The authors should cite newer, directly relevant studies to support their work.

Lines 63-65: microsatellite instability, k-ras and BRAF mutations should not be considered 'genetic disorders'. That term is used to describe heritable diseases.

Table 1 has interesting data - what does 'Grupy wiekowe badanych' (years old) mean? And if the study only considered subjects who smoked for at least 20 years, why would a row include <30 years old? Are there people who begin smoking at 10 years old who would have been consented for this study. Even 31-40 seems quite young, although more reasonable.

Figure 1 - not clear what the vertical lines indicate. Also, this type of plot is more typical of a survival curve. The histogram in Figure 2 seems to convey the same information and is more useful.

Why are no statistics indicated in Figures 2- 4? Data in Figure 4 is quite interesting.

In Figure 6, the y-axis is labeled as 'relative frequency'. What exactly does that mean? The frequency of developing a serrated polyp seemed to double in smokers vs. non-smokers. But what does the relative frequency actually imply in this data? Isn't the prevalence of serrated polyps ~ 5% in the general population?

Figure 7 is rather confusing as presented. Why are the 3 ACF groups not presented in each size group of adenomas, even if it is zero. And it may be better to combine the data in Figs 7 and 8 together for easier comparison.

Author Response

Our responses are presented in the file in the attachment
